# Detection of Del/Dup Inside *SHOX*/PAR1 Region in Children and Young Adults with Idiopathic Short Stature

**DOI:** 10.3390/genes12101546

**Published:** 2021-09-29

**Authors:** Jera Stritar, Lana Stavber, Maja Ficko, Primož Kotnik, Tadej Battelino, Katarina Trebušak Podkrajšek, Tinka Hovnik

**Affiliations:** 1Clinical Institute of Special Laboratory Diagnostics, University Children’s Hospital, UMC, 1000 Ljubljana, Slovenia; jera.stritar@kclj.si (J.S.); maja.ficko@kclj.si (M.F.); katarina.trebusakpodkrajsek@mf.uni-lj.si (K.T.P.); 2Department of Pediatric Endocrinology, Diabetes and Metabolic Diseases, University Children’s Hospital, UMC, 1000 Ljubljana, Slovenia; lana.stavber@kclj.si (L.S.); primoz.kotnik@kclj.si (P.K.); tadej.battelino@kclj.si (T.B.); 3Department of Pediatrics, Faculty of Medicine, University of Ljubljana, 1000 Ljubljana, Slovenia; 4Institute of Biochemistry and Molecular Genetics, Faculty of Medicine, University of Ljubljana, 1000 Ljubljana, Slovenia

**Keywords:** height, short stature, *SHOX*, MLPA, Sanger sequencing

## Abstract

Short stature is a common growth disorder defined as a body height two standard deviations (SD) or more below the mean for a given age, gender, and population. A large part of the cases remains unexplained and is referred to as having idiopathic short stature (ISS). One of the leading genetic causes of short stature is variants of short stature homeobox-containing gene (*SHOX*) and is considered to be responsible for 2–15% of ISS. We aimed to analyse the regulatory and coding region of *SHOX* in Slovenian children and young adults with ISS and to investigate the pathogenicity of detected variants. Our cohort included 75 children and young adults with ISS. Multiplex ligation-dependent probe amplification (MLPA) was performed in all participants for the detection of larger copy number variations (CNVs). Sanger sequencing was undertaken for the detection of point variants, small deletions, and insertions. A total of one deletion and two duplications were discovered using the MLPA technique. Only one of these four variants was identified as disease-causing and occurred in one individual, which represents 1.3% of the cohort. With Sanger sequencing, two variants were discovered, but none of them appeared to have a pathogenic effect on height. According to the results, in the Slovenian population of children and young adults with ISS, *SHOX* deficiency is less frequent than expected considering existing data from other populations.

## 1. Introduction

Short stature is a frequent developmental condition in childhood and is one of the most common reasons for visiting paediatric endocrinologists. The prevailing pathological causes of short stature are growth hormone (GH) deficiency, genetic disorders, hypothyroidism, and celiac disease. However, in 50–90% of the cases, no obvious pathological cause is found, so they are classified in the group of normal short stature–familial short stature, constitutional delay of growth and puberty, or idiopathic short stature (ISS). It is difficult to distinguish these three populations as they have multiple overlapping features; thus, they all could be considered to have idiopathic short stature. Short stature is also characteristic for infants born small for gestational age (SGA), the majority of which have catch-up growth in the first year of life [1,2].

According to the Consensus Statement from 2008, ISS is defined as a height two standard deviations (SD) or more below the corresponding mean height for a given age, sex, and population group without evidence of systemic, endocrine (e.g., GH deficiency, hypothyroidism, Cushing syndrome, precocious puberty), nutritional, chromosomal abnormalities (e.g., Turner syndrome), evident skeletal dysplasia, or other genetic syndromes (Noonan, Silver–Russell, Prader–Willi) [3]. ISS diagnosis is complex and based on medical history, physical examination, standard laboratory tests, and genetic analysis. There is still a large proportion of those in whom the specific genetic cause by standard and existing genetic analysis remains undetermined.

Processes involved in growth can be interrupted by genetic, epigenetic, and environmental factors [4]. Moreover, it is well known that height is one of the human polygenic traits with the highest degree of heritability (60–80%) [1]. Genome-wide association studies (GWAS) revealed that the accumulation of multiple genetic variants (i.e., single nucleotide polymorphisms (SNP)) affects growth, but each contributes a very small proportion to a final height [4]. However, in some individuals reduced stature is caused by a specific genetic change with a large effect, such as variants in the regulatory and coding region of the short stature homeobox-containing gene (*SHOX*) that are supposedly responsible for 2–15% of short stature, formerly diagnosed as idiopathic [1,5,6,7].

The *SHOX* gene is located in the telomeric pseudoautosomal region 1 (PAR1) on the short arm of sex chromosomes and plays an important role in the skeletal development of vertebrates. It escapes the inactivation of the X chromosome and consequently remains expressed on both sex chromosomes. Even though the *SHOX* gene is located on sex chromosomes, it is inherited in an autosomal dominant manner [6,8]. The most common genetic variants in the *SHOX* gene are deletions, followed by duplications and point variants [9].

*SHOX* haploinsufficiency is expressed by a variable clinical phenotype, from extremely short stature to mild short stature with no additional clinical features [10]. The most severe phenotype manifests as Langer mesomelic dysplasia, where both copies of *SHOX* are lost. Skeletal deformity with severe short stature, mesomelia, and Madelung deformity is characteristic. A milder form of skeletal dysplasia is present in Leri–Weill dyschondrosteosis (LWD), resulting from a heterozygous variant in the *SHOX* gene. The same phenotype is expressed in Turner syndrome, where the entire X chromosome is lost along with one copy of the *SHOX* gene. Heterozygous variants of *SHOX* were also found in individuals with nonsyndromic short stature [11].

This study aimed to assess the frequency of genetic variants in the regulatory and coding regions of the *SHOX* gene in a group of children and young adults with ISS. In our cohort study, a mutational analysis was performed by multiplex ligation-dependent probe amplification (MLPA) for detecting larger copy number variations (CNVs), followed by Sanger sequencing to determine point variants, small deletions, and insertions.

## 2. Materials and Methods

### 2.1. Participants

The cohort included 75 subjects (39 female, 36 male), aged 2 to 22 years old, referred to the University Children’s Hospital of Ljubljana between 2015 and 2019. All participants were selected following strict inclusion criteria (i.e., height below −2 SD) and exclusion criteria—GH deficiency, chronic systemic diseases (cystic fibrosis, celiac disease, chronic inflammatory bowel disease, type 1 diabetes, etc.), endocrine disorders (Cushing’s syndrome, hypothyroidism), defined skeletal dysplasia, chromosomal abnormalities (Turner syndrome, Down syndrome) and growth influencing medications (glucocorticoids) [12]. Arginine and L-Dopa GH stimulation tests were performed according to previously published test procedures [13]. Serum GH levels were determined by immunoassay using Immulite 2000 (Siemens). Bone age was evaluated based on Greulich and Pyle Atlas (GP) bone age determination system, 2nd edition, or determined with the BoneXpert program [14]. Z-scores for height were calculated using the LMS method (L (curve Box-Cox), M (curve median) and S (curve coefficient of variation)) and the British 1990 reference growth data [15]. Detailed patients’ features are given in Appendix A.

The study was conducted according to the guidelines of the Declaration of Helsinki and approved by the Slovene Medical Ethics Committee (0120-36/2019/4). All participants or their legal guardians provided written informed consent before the study.

### 2.2. Methods

Whole blood EDTA samples were collected. Isolation of genomic DNA was performed using a commercial FlexiGene DNA isolation kit (Qiagen, Hilden, Germany) according to the manufacturer’s instructions [16]. The concentration of the isolated DNA was determined using a UV-VIS spectrophotometer. Samples of isolated DNA were stored at 4 °C throughout the study. All samples were first analysed by the MLPA assay to discover larger CNVs inside the coding sequence of the *SHOX* gene and enhancer regions in PAR1. Samples in which no change was detected by the MLPA method were further analysed by sequencing all exons in the *SHOX* gene coding region.

MLPA analysis was performed using the commercial SALSA MLPA Probemix P018-G2 SHOX kit (MRC-Holland, Amsterdam, The Netherlands) according to the manufacturer’s instructions. It contains a total of 48 MLPA probes with amplification products between 124 and 504 nucleotides. Thirty-two of these probes are for the PAR1 region on chromosome Xp22/Yp11. Several other probes are included for *SHOX* regulatory regions upstream and downstream of the gene, and 13 flanking probes in the probemix are targeting the X chromosome outside the *SHOX* area. In addition, probemix also contains nine reference probes and ten quality control fragments [17,18]. Polymerase chain reaction (PCR) fragments were separated by genetic analyser ABI 3500 (Applied Biosystems, Waltham, MA, USA). The MLPA results were analysed using Coffalyser.Net software [19]. No DNA control and at least three reference DNA samples were included in each MLPA reaction. The results were presented as probe ratios, with the normal number of copies expressed as the ratio of 0.80–1.20, whereas deletions and duplications were expressed as a ratio less than 0.65 (loss) or greater than 1.30 (gain), respectively. Each detected alternation was confirmed by an independent MLPA reaction.

In Sanger sequencing analysis the *SHOX* coding region was amplified by PCR, using in-house designed sets of primers. Amplification was performed with an initial step at 95 °C for 2 min, followed by 35 cycles at 95 °C for 30 s, 64 °C for 30 s and 72 °C for 40 s, and the final extension at 72 °C for 7 min. After agarose electrophoresis and purification of the amplicons, sequence analysis was performed using BigDye Terminator v3.1 Cycle Sequencing Kit, followed by capillary electrophoresis with 3500 Genetic Analyser (Applied Biosystems, Waltham, MA, USA). The final results were analysed using the BLAST program from the NCBI browser [20], which shows any discrepancies between the sample and control reference sequences. The first exon of the *SHOX* gene is noncoding; therefore, it was not included in the sequence analysis.

## 3. Results

Our study group comprised 75 participants (52% female) with a mean age of 10.6 years. The mean height was –2.8 SD for girls and −2.6 SD for boys. Subtle dysmorphic features that were not assigned to any known syndrome (isolated clinodactyly, micrognathia, hypertelorism, broad nasal bridge, epicanthus, low-set ears) were present in 40 subjects. Six girls additionally had disproportion of extremities (slight leg length inequality, shorter forearms, small fifth finger unilaterally). Bone age (BA) compared to the chronological age was delayed in 53% and advanced in 13% of the participants. The final height of both parents was taken into account as well. Mid-parental height (MPH) was calculated using EBMcalc Medical Calculator [21]. At the time of the study, eight participants were already treated with recombinant human growth hormone (rh-GH). For these individuals, data prior to the treatment were used. None of the participants presented with the Madelung deformity. Details are given in Appendix A.

We identified altogether five alternations in the *SHOX* gene region including non-coding enhancer regions, of which one was present in three different subjects. Three alternations were detected by MLPA: long heterozygous deletion at least 1340 kbp in size, long heterozygous duplication at least 324 kbp in size, and a heterozygous duplication of the *SHOX* CNE-5 enhancer element. Sanger sequencing revealed two different heterozygous alternations: a silent point variant in the exon 2 (NM_000451.3:c.63C > T) present in three different subjects and deletion of three codons in the noncoding region between exons 4 and 5 (NM_000451.3:c.544+15_544+23del).

### 3.1. Copy Number Variants Identified by MLPA Assay

The complete heterozygous *SHOX* deletion, encompassing the entire coding region together with all highly conserved non-coding DNA elements (CNEs) on both sides of the gene (except CNE-5), as well as four other genes (*CRLF2*, *CSF2RA*, *IL3RA*, and *ASMT*), was present in participant 42 (P42). The deletion is located on the Xp22-PAR1 chromosome region spanning from X:000,380638 to X:001,712090. The length of the deletion was estimated by summing the distances between the individual probes with only one copy (probe ratio 0.5), showing a deletion of approximately 1340 kbp.

At least 324 kbp long heterozygous duplication, encompassing all exons and multiple regulatory elements in the *SHOX* region, was found in participant 75 (P75). Complete *SHOX* duplications appear to be extremely rare and also present in clinical conditions unrelated to ISS [22,23].

Isolated heterozygous duplication of the CNE-5 enhancer element was identified in one female participant (P47). The duplication measured at least 364 bp in length and did not include any *SHOX* coding sequences.

### 3.2. Sequence Analysis of SHOX Gene

Two different heterozygous variations were found with gene sequencing, one of which was present in three participants. According to the GnomAD population database, the frequency of silent variant M_000451.3:c.63C > T in the general population is 0.4% and was classified as benign according to the recommendation of the American College of Medical Genetics (ACMG) [24]. The frequency of the intronic deletion NM_000451.3:c.544+15_544+23del is 0.0008% and was classified as a variant of unknown significance according to the recommendation of the ACMG. Neither of these two variations affects the amino acid sequence in the protein; they are located in regions that are less important for intron excision and are also present in the general population. Therefore, it is very likely that these two variations are not disease-causing.

No CNVs, point variants, or smaller deletions/duplications were identified in the other 68 study participants.

## 4. Discussion

Determining the genetic cause of short stature in growing children is important from several points of view. It enables us to better predict a natural outcome based on the published cases (e.g., final height and possible concomitant diseases) and suggests the best treatment options (e.g., use of recombinant human growth hormone (rh-GH) in children with *SHOX* variations) [8].

The usual approach for detecting disease-causing variations in the *SHOX* region begins with determining CNVs, especially larger deletions, as these are the most common. Point variants and small deletions or insertions are less common. The incidence of *SHOX* gene variants in children with ISS is currently estimated at 2–15% based on a number of screening studies [10,25,26]. With new, more sensitive genetic tests (e.g., MLPA) and the discovery of the importance of amplification sequences, this estimate may increase. MLPA enables the simultaneous analysis of CNVs in broad genomic regions and allows the analysis and comparison of several patients in the same experiment. It is fast, easy, and very sensitive to detect larger deletions in the gene itself and in its remote regulatory regions as well [6,27]. For a comprehensive diagnosis of ISS, it is important to combine different methods, from cytogenetic to molecular genetics, that enable the identification of all types of variations. In this study, we aimed to determine the frequency of genetic variations in the *SHOX* gene region in a group of 75 Slovenian children and young adults with ISS with two independent methods—MLPA to determine CNVs, and Sanger sequencing to identify point variants, small deletions, and insertions.

The most clinically important finding of our study was a 1340 kbp long deletion of the entire *SHOX* gene present in a boy with short stature without typical LWD clinical signs. The participant showed no signs of dysmorphism or body disproportion; however, shorter extremities and Madelung deformity were present in his mother. The mentioned deletion of the entire gene is the only undoubtedly causal variant identified in our group of short stature participants. To accurately define the breaking points of the deletion and its exact length, sequencing of the entire region should be performed. However, it is clear that the identified deletion covers the entire coding region of the *SHOX* gene together with all CNEs, except CNE-5, as well as four other genes (*CRLF2*, *CSF2RA*, *IL3RA,* and *ASMT*).

In addition to the 1340 kbp long deletion, we detected two other variants with the MLPA method—at least 324 kbp long duplication in the *SHOX* region and isolated duplication of the CNE-5 enhancer. In the study of *SHOX*/PAR1 duplications, Benito-Sanz et al. noted the presence of *SHOX* duplications in other clinical conditions as well, so their pathogenicity in short stature is questionable [5]. Duplication of the CNE-5 enhancer appears in the literature in combination with the deletion of the CNE-3 enhancer, which the author considers as a causal variant in the family with ISS [28]. However, there is not enough evidence for stand-alone CNE-5 duplication to be pathogenic, and further research should be undertaken. Therefore, both variants described above are considered benign.

Among the above-mentioned alternations, only complete *SHOX* deletion was identified as disease-causing, which represents 1.3% of our study group. Our findings therefore slightly deviate from the data from the literature, which states a 2–15% incidence of *SHOX* variations among individuals with ISS [10,25,26]. The outcome of the analysis is significantly influenced by the formulation of inclusion and exclusion criteria. The most basic criteria for creating a cohort group, also used in our study, are growth retardation (height below –2 SD) and no specific known causative disorders. Differences between the criteria occur in the inclusion/exclusion of patients with dysmorphic signs and also regarding the age of the patients since some authors include only prepubertal children, while others include persons up to 29 years of age. The correct interpretation of the clinical definition of short stature is crucial and enables the most homogeneous study group. Diversity in diagnostic yield is also due to the use of various methodological approaches in studies used to detect different types of variants. The changes may thus be located in regions that are not visible with the approach used. Nevertheless, we carefully selected a homogenous group of patients with ISS and applied all appropriate genetic tests to detect causal variants inside the *SHOX* gene and enhancer region.

In addition to variants in the *SHOX* gene itself, enhancer sequences are also of great importance and, when altered, they may contribute to the development of growth failure. This was demonstrated by Chen et al. in their study of 735 participants with ISS and 58 Leri–Weill syndrome patients using the FISH and MLPA methods. In the ISS group, the presence of CNV in the *SHOX* region was confirmed in 31 individuals (4.2%), 8 of whom (26%) carried the microdeletion only in enhancer sequences positioned at least 150 kb away from the coding region. In Leri–Weill syndrome patients, 29 microdeletions were identified, 13 of which (45%) involved only enhancer sequences. In all these cases, the *SHOX* gene remained intact [10]. In the cohort of French children with ISS, Rosilio et al. identified *SHOX* deficiency in 16.9%, more than half of which had a deletion downstream of the gene in the enhancer region [7]. This confirms that enhancer deletions in the *SHOX* region are a relatively common cause of short stature in children diagnosed with ISS. In this region, a total of seven conserved non-coding elements have been identified that act as enhancer elements for the *SHOX* gene transcription. Four of them (CNE4, CNE5, ECR1, CNE9/ECS4) are located downstream of the gene, and three of them (CNE-5, CNE-3, CNE-2) upstream of the gene. The loss of enhancer elements leads to reduced gene expression and consequently affects the growth and development of the individual [28,29].

The results of this research are consistent with a similar study in the Slovenian population conducted by Hovnik et al. in 2015 at the University Children’s Hospital of Ljubljana. The *SHOX* region was analysed by fluorescence in situ hybridization to identify large deletions, and direct DNA sequencing was used to identify point variants and small deletions or insertions. They concluded that small or large-scale *SHOX* alternations are not a common cause of short stature among Slovenian children with ISS, as they did not find any variations in the *SHOX* gene in a group of 40 participants of the study [27].

In conclusion, despite the integrated genetic approach, using MLPA analysis and Sanger sequencing, our study results indicate that *SHOX* variants are not a common cause of ISS in Slovenian patients. Further research and functional studies are needed to confirm the pathogenicity of different SHOX variants, especially duplications. A definite genetic diagnosis also enables a more accurate prognosis and the possibility of treatment. In prepubertal children with ISS due to *SHOX* deficiency, an increase in final height by 7 to 10 cm can be achieved by rh-GH therapy [30].

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
