# Peer review of "Detection of Del/Dup Inside SHOX/PAR1 Region in Children and Young Adults with Idiopathic Short Stature"

_genes, 2021, doi:10.3390/genes12101546_

Round 1

Reviewer 1 Report

Very clear and good article about possible genetic causes of short stature. In the heading of Materials and Methods is mentioned that excluded were all children with growth influencing medications like glucocorticoids. I wonder if you have controlled for corticosteroids used before birth. Sometimes rather high amounts of steroids are given to try to prevent hyaline membrane disease in mothers with a threatening premature labour.  So have you controlled for prematurity or growth retardation before birth? This is only a minor remark.

The study gives results that in their cohort alterations in the SHOX region are not a common cause of short stature,

It is a nice very good readable article.

Reviewer 2 Report

Thank you for the opportunity to review the manuscript “Detection of del/dup inside SHOX/PAR1 region in children and adolescents with idiopathic short stature” by Jera Stritar et al. The Authors conducted the study of a rare genetic reason for short stature (SS).

However, there are some points that need to be improved.

  1. Line 76-77: the patients’ cohort is described as “aged from 2 to 22 years old”, therefore we cannot describe the study group as “children and adolescents”. They are rather children and young adults.
  2. Line 37-40: it seems that the sentence is incomplete (“…”). Moreover, if we describe the genetic causes of SS it would be worthwhile adding some more important causes, not only Turner syndrome (e.g. Noonan, Silver-Russell, Prader-Willi syndrome).
  3. There is lack of discussion (Introduction, Discussion) regarding the most common clinical diagnosis in SS: delayed growth and puberty and familial SS (of course there may be a genetic, unknown reason as a background of the clinical diagnosis). Furthermore, children small for gestational age (SGA) or with intrauterine growth retardation (IUGR) are not mentioned.
  4. Line 46: “GWAS studies”- explain the abbreviation
  5. Line 78-83:
    1. Please give the more detailed information about the inclusion and exclusion criteria, especially the endocrine work-up, especially the criteria for GHD exclusion (e.g. in a supplementary file).
    2. Was karyotype performed in all girls/women, as recommended by paediatric endocrinology societies?
    3. What were the neonatal characteristics, were there children with SGA/IUGR within the cohort?
  6. Lines 123-124:
    1. Please give more details on “mild dysmorphic features”.
    2. “7 girls additionally had disproportion of extremities compared to the trunk”- describe these disproportions more precise.
  7. Lines 125-127: “Bone age (BA) compared to the chronological age was delayed in 52% and advanced in 18% of the participants.”- I would suggest to be more precise and give e.g. chronological age/bone age (CA/BA)

  1. Lines 127-128: “The final height of both parents was taken into account as well. 36/84 (42.9%) individuals had at least one parent with slightly lower body height, of which 8/84 (9.5%) with markedly short stature.”- again in my opinion these data must be more precise (e.g. was it mother or father, what was their height SDS, what was the mean parental height SDS (mph SDS) in the whole cohort; comparison of patient’s height SDS to mph SDS). It could be also included as a supplementary file.
  2. I found the discussion interesting and valuable. However, there is lack of possible treatment discussion, it is only mentioned in the final conclusion.
  3. Line 260: it should be “written informed consent”.

Round 2

Reviewer 2 Report

Please see the attached review.
